# Genome-Wide Association Mapping of Prostrate/Erect Growth Habit in Winter Durum Wheat

**DOI:** 10.3390/ijms21020394

**Published:** 2020-01-08

**Authors:** Daniela Marone, Monica Rodriguez, Sergio Saia, Roberto Papa, Domenico Rau, Ivano Pecorella, Giovanni Laidò, Nicola Pecchioni, Julia Lafferty, Matthias Rapp, Friedrich H. Longin, Pasquale De Vita

**Affiliations:** 1Research Centre for Cereal and Industrial Crops, CREA, SS 673, km 25.200, 71122 Foggia, Italy; daniela.marone@crea.gov.it (D.M.); sergio.saia@crea.gov.it (S.S.); ivano.pecorella@gmail.com (I.P.); Giovanni.lai79@libero.it (G.L.); nicola.pecchioni@crea.gov.it (N.P.); 2Department of Agriculture, University of Sassari, Via E. de Nicola, 14, 07100 Sassari, Italy; mrodrig@uniss.it (M.R.); dmrau@uniss.it (D.R.); 3Centro per la Conservazione e Valorizzazione della Biodiversità Vegetale, Università degli Studi di Sassari, SS 127bis, km 28.500 Surigheddu, 07041 Alghero, Italy; 4Research Centre for Cereal and Industrial Crops, CREA, SS 11, km 2.500, 13100 Vercelli, Italy; 5Department of Agricultural, Food and Environmental Sciences, Università Politecnica delle Marche, Via Brecce Bianche, 60131 Ancona, Italy; rpapa@univpm.it; 6Saatzucht Donau GesmbH & CoKG, Saatzuchtstrasse 11, A-2301 Probstdorf, Austria; julia.lafferty@saatzucht-donau.at; 7State Plant Breeding Institute, University of Hohenheim, Fruwirthstraße 21, 70593 Stuttgart, Germany; Matthias_Rapp@uni-hohenheim.de (M.R.); Friedrich.Longin@uni-hohenheim.de (F.H.L.)

**Keywords:** durum wheat, juvenile growth habit, tiller angle, QTL, candidate gene

## Abstract

By selecting for prostrate growth habit of the juvenile phase of the cycle, durum wheat cultivars could be developed with improved competitive ability against weeds, and better soil coverage to reduce the soil water lost by evaporation. A panel of 184 durum wheat (*Triticum turgidum* subsp. *durum*) genotypes, previously genotyped with DArT-seq markers, was used to perform association mapping analysis of prostrate/erect growth habit trait and to identify candidate genes. Phenotypic data of plant growth habit were recorded during three consecutive growing seasons (2014–2016), two different growth conditions (field trial and greenhouse) and two sowing periods (autumn and spring). Genome-wide association study revealed significant marker-trait associations, twelve of which were specific for a single environment/year, 4 consistent in two environments, and two MTAs for the LSmeans were identified across all environments, on chromosomes 2B and 5A. The co-localization of some MTAs identified in this study with known vernalization and photoperiod genes demonstrated that the sensitivity to vernalization and photoperiod response are actually not only key components of spring/winter growth habit, but they play also an important role in defining the magnitude of the tiller angle during the tillering stage. Many zinc-finger transcription factors, such as C2H2 or CCCH-domain zinc finger proteins, known to be involved in plant growth habit and in leaf angle regulation were found as among the most likely candidate genes. The highest numbers of candidate genes putatively related to the trait were found on chromosomes 3A, 4B, 5A and 6A. Moreover, a bioinformatic approach has been considered to search for functional ortholog genes in wheat by using the sequence of rice and barley tiller angle-related genes. The information generated could be used to improve the understanding of the mechanisms that regulate the prostrate/erect growth habit in wheat and the adaptive potential of durum wheat under resource-limited environmental conditions.

## 1. Introduction

Plant architecture integrates a set of important agronomic traits in wheat, such as plant height, tiller number, juvenile growth habit, flag leaf angle and spike characteristics, that are important for increasing crop yield potential. Plant breeders have extensively modulated such architectural traits: the extraordinary increase in wheat yield that has been registered with the introduction of modern semi-dwarf wheat varieties during green revolution was partially due to the improvement of the plant architecture [1], e.g., in terms of plant height [2] and flag leaf angle [3].

Durum wheat (*Triticum turgidum* subsp. *durum*) is grown on 8 to 10% of all the wheat cultivated area in the world and is an economically important crop in the Mediterranean area where it represents one of the most important agricultural crops. Changes in global average temperature, precipitation regime and increase of atmospheric CO_2_ concentration will impact the crop productions at various rates in different parts of the world, particularly in those areas, such as the Mediterranean basin, already considered as one of the most critical and vulnerable geographic zones [4]. Therefore, it is important to increase the knowledge of durum wheat plant architecture, in terms of size, shape and orientation of the shoot, to improve the yield performances of durum wheat under the effect of ongoing climate change [5] and increase its ability to rapidly cover the soil, that is directly related to both yield and competitivity against weeds [6].

Tillering, or the production of lateral branches (i.e., culms), is a key component of yield for straw cereals such as rice, barley, bread and durum wheat ([7,8] and reference therein), given that the number of tillers affects the number of fertile spikes and consequently the kernel numbers per unit area. So, selecting cultivars with moderate-to-high tillering ability represents an important breeding objective. However, for monocotyledonous crops, the dynamics of tiller angle from vertical, during the juvenile growth stage (i.e., from prostrate to semi-prostrate and erect growth habit) is another important agronomic trait that should be considered, because a dense ground cover affects the interception of light for photosynthetic accumulation, the inhibition of weed growth and the reduction of water evaporation from soil [9].

Rice varieties show large variation in tiller angle, as a complex quantitative trait, the genetic bases of which have been extensively investigated using quantitative trait loci (QTL) analysis [10,11,12]. Several underlying genes, including *LAZY1* (*LA1*), *PROG1* (Prostrate Growth 1), *DWARF4*, and *TAC1* (Tiller Angle Control 1), associated with tiller angle, have been cloned through map-based cloning, and the molecular basis has been clearly elucidated [13,14,15,16].

In barley, plants possessing the semi-dwarfing *sdw1*/*denso* gene are characterized by prostrate growth, whereas plants with its dominant allele are characterized by erect growth providing an effective morphological marker of this gene. Further analysis revealed that barley *sdw*/*denso* gene, an ortholog of the rice *SD1* gene [17], is located on chromosome 3H based on the barley reference genome sequence and has a pleiotropic effect on several agronomic traits such as plant height, heading and flowering time [18]. Other QTLs that determined the prostrate/erect growth habit in *Hordeum spontaneum* have been found on the long arm of chromosomes 1H, 3H and 6H, with that on chromosome 3H located in the same region of *sdw1*/*denso* gene *locus* [19].

Studies carried out on wheat are instead very limited. In the past, Li et al. [20] found few specific QTLs for juvenile growth habit in wheat, as a quantitative trait associated with other morphological traits. In particular, they found three regions on 6AS, 1DS and 2DS controlling tiller number that also influenced prostrate/erect growth habit. More recently, a genomic wide association study (GWAS) has been conducted for seedling habit, together with other important agronomic traits, and significant marker-trait associations (MTAs) have been found for this trait on chromosomes 3B, 4A and 6B [21]. Significant MTAs for the prostrate/erect trait have also been identified in durum wheat: Giraldo et al. [22] found two chromosome regions on 3A and 3BL, significantly associated to plant juvenile growth habit in tetraploid wheats (*T. durum* and *T. dicoccum*). In the case of barley, the close association existing between the semi-dwarfing *sdw1*/*denso* gene regulating plant height and the prostrate growth habit could be explained by the pleiotropic effect of *sdw1*/*denso* gene on tiller angle [23].

The prostrate/erect growth type, as flowering, is known to be highly influenced by environmental conditions such as temperature and day-length [24]. This also means that adaptation genes such as *Vrn* and *Ppd* could influence the expression of juvenile growth habit trait either because in linkage, or because exert a pleiotropic effect on this trait [25]. Also freezing tolerance coupled to winter growth habit was found to be associated with prostrate growth type in wheat; a gene controlling prostrate growth was found to be closely linked with *Fr1*-*Vrn1* locus on chromosome 5A in the pioneering study of Roberts [26] about the genetic control of such traits. However, prostrate growth type can also be found in cultivars with low vernalization requirements but high photoperiod response, indicating since the early studies that sensitivity to vernalization and photoperiod are the two major components associated to tiller angle from vertical during the juvenile growth stage in wheat [27]. 

In the light of these considerations, it is therefore useful to dissect the genetic and molecular bases underlying the plant juvenile growth habit in terms of tiller angle, possibly in relation to major adaptation loci, being this trait very important in determining plant architecture and influencing yield component traits. The rapid advances in genotyping technologies enabled genotyping-by-sequencing (GBS) also in durum wheat, generating a density of genome-wide markers. The GBS platform of DArT-seq (Diversity Array Technology DArT, Canberra, Australia) allows for the selection of genome fractions that predominantly correspond to active genes. Association mapping (AM), also known as linkage disequilibrium (LD) mapping, is a powerful and promising tool for gene detection in crop plants [28]. Different traits have been genetically dissected in tetraploid wheat by means of this approach, such as root traits [29,30,31,32,33], resistance against stem rust, leaf rust and stripe rust [21,34,35,36], resistance to Fusarium Head Bligth (FHB) [37], various agronomic traits, biomass and yield components [38,39,40], and yellow pigment content [41].

With the longer-term aim of selecting genotypes with a desirable plant architecture to be used in durum wheat breeding, the aim of this study was to perform association mapping for prostrate/erect growth habit trait in a panel of 170 diverse winter and 14 spring durum wheat genotypes, previously genotyped with DArT-seq markers [42], and to identify candidate genes based on available sequences of markers located at the MTAs, in order to provide valuable information for better understanding the genetic mechanism of the tiller angle trait in wheat. Moreover, a bioinformatic approach has been pursued, to search for functional homologous genes in wheat by using the sequence of genes previously identified in other cereal species and found to be associated with tiller angle, namely *PROG1*, *TAC1*, *LAZY1* and *SD1* in rice. 

## 2. Results

### 2.1. Phenotypic Variation for the Trait

The statistical parameters of the plant growth habit for each environment are shown in Table 1 and Table 2. Phenotypic means ranged from 3.46 in FF14 (the first letter of the abbreviation of each experiment is for the sites, the second is for the kind of experiment and the number is for the cropping season, see Section 4.2 for a complete explanation. FF14 is for Foggia in the field in 2014) to 6.90 in HF14 (Hohenheim in the field 2014), with a grand mean of 5.06. All distributions were slightly platykurtic, except for PF13 (Probstdorf in the field 2013) and genotypic means (expressed as G LS means) that showed a relatively low kurtosys (−1.78 and −1.50, respectively). Broad sense heritability was very high and ranged from 0.79 in PF13 to 0.99 in FG15 (Foggia in the Greenhouse in 2015), indicating a tight genetic control (Table 1). Genotypes and Genotypes × Environment (G × E) were significant at *p* ≤ 0.001, making possible analyses for single environment (Table 2). Best linear unbiased predictions (BLUPs) were then used to confirm results of marker-trait associations. Skewness of the habit distribution in all environments, G mean, and G × E distributions were scarcely left-skewed, except for the environment HF14, which was moderately left-skewed (Figure 1). Mean values across locations, LSmeans (both across locations and by genotype × environment) and, correlations (*r*) coefficients calculated for determining the relations for the average values of the various environments, are given in Appendix A. 

### 2.2. Population Structure and Association Mapping

The population structure determined on the 184 durum wheat genotypes by means of the 30,611 DArT-seq markers by STRUCTURE and DAPC approaches (Appendix A), both evidenced the presence of four genetic groups named G1, G2, G3 and G4, in addition to the admixed group. It clearly showed that each group was different from others and that group G3 contains almost all spring genotypes, mixed with winter types (Appendix A). 

A high correlation between results from the two methods was found when considering the attribution of the different genotypes to each group (χ^2^ = 386.001, *p* < 0.0001). 

Genome-wide association analysis of the prostrate/erect growth habit at the tillering stage evidenced different MTAs for the LSmeans across all environments (Figure 2), as for the mean values of each single environment (Figure 3). The results of the genome scans for the tiller angle/juvenile growth habit trait were summarized in Table 3, Figure 2 and Figure 3, where only significant MTAs (above Bonferrroni threshold) are highlighted. Q-Q plots in Figure 2 and Figure 3 showed as the model well fitted the data, with observed values (markers/dots) being very close to the predicted values (straight line).

When LSmeans from all the environments were investigated, only two MTAs were detected, one on chromosome 2B (D1202558), and the second hit was among the unmapped markers (D2277949), explaining 16% and 21% of phenotypic variance, respectively (Table 3, Figure 2).

Considering the single environments, a total of 27 MTAs was detected for prostrate/erect habit on all chromosomes, except for chromosome 1A, 1B and 5B. Some markers were found to be associated to the trait when evaluated in more than one site/year, and/or when considering the LSmean values. In particular, D1202558 marker on chromosome 2B was in common to FF14, FF15 (two different years and sowing periods) and LSmeans values, and D1665929 on 4A was found in FF14 and FG15, both characterized by a spring sowing time. The detected QTLs were represented by single markers and only the regions identified on chromosomes 5A (for HF14 and FG15), 2B (for FF14, FF15, PF13 and LSmeans), and 4B (for HF14 and FG15) were found associated to a set of closely linked markers (Table 3). Nevertheless, each QTL-tagging marker was co-mapping with many other DArT-seq and DArT markers, according to the consensus wheat map version 4.0 available on Triticarte website (https://www.diversityarrays.com/technology-and-resources/genetic-maps/). In particular, single MTAs were detected in single environments on chromosomes 2A (FF16), 2B (HF14), 3A (PF13), 6A (FF14), 6B (HF14), and 7B (PF13). Moreover, two distinct regions from different environments were identified on chromosomes 2B, 3B, 4A, 4B and 7A. Finally, a group of nine markers for which no map information was available, showed association to the growth habit, all identified in a specific environment. 

The unmapped MTAs were conditionally located on the genome by means of LD between the unmapped and mapped DArT-seq markers positioned on the wheat consensus map. The LD was previously assessed [42] using 30,475 DArT-seq markers mapped on the wheat consensus map available on Triticarte website, using PLINK 1.07 (http://pngu.mgh.harvard.edu/purcell/plink/; [43]). Taking into account the LD decay estimated by Sieber et al. [42] a threshold around 2–5 cM within chromosome, was considered. On the basis of these results, we can putatively locate the MTAs D4004513 and D1744736, both significant in HF14, on the chromosome 6B in complete LD (r^2^ = 1) with S2258653 (41.5 cM) located in the same region where the MTA D2289020 has been identified (36.8 cM). The marker D3946194 was in LD with a group of markers such as D3024894, also positioned on 6B but in a different region (65 cM) with respect to the MTA D2289020. Other unmapped MTAs can be putatively located in regions already targeted in this work, such as D2277949 which is in LD with D2276320 (mapped on 5A at 167.7 cM), and D3935715 and D3533805 both in LD with the MTA D2295851 (mapped on 7A at 32.3 cM). Finally, the marker S984195 was in LD with S1065494 (mapped on 7B at 95.8 cM) which is very far from the other MTA here identified on the same chromosome (D1112046 at 184.4 cM). Therefore, two new regions (D3946194 and S984195 putatively located at 65 and 95.8 cM on chromosome 6B and 7B, respectively) can be suggested as likely associated to the prostrate/erect growth habit, whereas any genetic position has been assumed for the two remaining unmapped MTAs (D3944539 and S1218298). 

Interestingly, when we searched for any association between the MTAs identified in this work and known genes from literature (i.e., *Vrn*, *Ppd* and *Rht*), we found two DArT-seq (D1202558 and D1031337) that mapped on chromosomes 2B (at 62.3 cM) and 7A (at 91.8 cM) respectively, in the same region where *Ppd-B1* (based on marker wPt-7695 at 62.39 cM) and *Vrn-A3* (wPt-9314 at 89.52 cM) were located, according to common markers reported by Maccaferri et al. [44] and Le Gouis et al. [45]. The MTAs identified by the markers D1395268 and D1720107 on 4B (at 133.6 and 138.3 cM, respectively) could correspond to the gene *Vrn-B2*, that was mapped close to wPt-5265 (at 148.34 cM) as reported by Le Gouis et al. [45]. Comparing the position of this DArT marker with that of common markers (wPt-3608, IWA5358 and Xbarc193) reported in other maps [44,46], the region could be the same of or very close to *Vrn-B2*. Moreover, the second region on 4BS associated to plant growth habit was verified for correspondence to the dwarfing gene *Rht-B1*. As reported by He et al. [47], *Rht-B1* mapped very close to the DArT-seq D3064743, that was at 50 cM far from the MTA D1110414 here identified on 4BS according to the wheat consensus map on Triticarte website. Finally, the region identified on chromosome 5A by multiple associations corresponded to the gene *Vrn-A1*, mapped on a durum wheat linkage map constructed by using the same DArT-seq array used to genotype our collection [8].

Considering that several MTAs for growth habit mapped on chromosome 5A near to the vernalization gene, as also the unmapped MTA D2277949, significant in all environments, putatively locate on chromosome 5A at 167.7 cM (based on the LD with D2276320), we gained an insight into this region to better understand its role in the expression of both frost tolerance and growth habit. Results from GWAS using mixed linear model (MLM) showed a main association peak for frost tolerance trait in a range between 106.91 and 114.89 with the top single-nucleotide polymorphism (SNP) (D1111190, *p* = 5.34 × 10^−13^) located at 111.66 cM (Figure 4A). The association peak for tiller angle was located between 164.32 and 168.65 cM with its top SNP (D2276320, *p* = 8.68 × 10^−10^) at 167.75 cM (Figure 4B). A second smaller peak that did not reach the significance (*p* = 7.54 × 10^−6^) was visible on the left with a top SNP at 111.66 cM. We then decided to use the top SNP for frost tolerance as a cofactor in a second MLM association analysis for growth habit recorded at Hohenheim (HF14). The results show that the main peak for growth habit is still present at 167.75 cM with a strong reduction of the smaller peak at 111.66 cM (Figure 4C). This suggested that two different regions were likely implicated in the expression of these two traits and that the MTA D1111190 correspond to the locus *Fr-A2* on chromosome 5A as previously reported by Sieber et al. [42]. 

### 2.3. Candidate genes

The Confidence Intervals (CIs) including MTAs, calculated according to the LD decay (5 cM), were from 1.2 to 6 cM based on the availability of marker sequences in those regions. Table 4 and Table 5, respectively, reported the physical intervals retrieved from the genome assemblies of the *T. dicoccoides* accession Zavitan and of the durum wheat cv Svevo corresponding to the genetic ones, the number of the annotated genes within these intervals, together with the number of related-growth habit genes, as previously described in literature. The physical intervals are very similar in both genomes in terms of size, except for chromosomes 3A, 3B, 4B, 5A and 6B, whereas the number of annotated genes in the intervals varied, particularly from 35 on 7A and 7B to 140 on 5A on the Zavitan genome, and from 33 on 4B to 461 on 4A on the durum wheat genome. Generally, a higher number of genes have been found in the Svevo genome, except for the region on 4B (132.4–138 cM) for which the reverse was true. Many zinc-finger transcription factors were found on each chromosome region considered in both genomes, such as C2H2 or CCCH-domain zinc finger proteins, known to be involved in plant growth habit and in leaf angle regulation, but also ethylene-responsive transcription factors, gibberellin-regulated family proteins, MADS-box factors and genes affecting plant growth regulators (Table 4 and Table 5; Appendix A). The higher numbers of trait-related genes, considering both genomes, were found on chromosomes 2A, 2B, 3B, 4A, 4B, 5A, 6A and 6B. All annotations were fully described in Appendix A, also including disease resistance proteins, factors affecting cold acclimation, kinases, transport receptors, sugar transporters, different kinds of transcription factors and genes encoding signal transduction pathway proteins. 

### 2.4. Search for Orthologs of the Rice Genes PROG1, LAZY1, TAC1 and SD1

Results of BLAST search against the ‘Zavitan’ and ‘Svevo’ transcripts by using the rice protein sequences of *PROG1*, *TAC1*, *LAZY1* and *SD1* are summarized in Appendix A. Significant hits on chromosomes 4B, 5A, 5B and 7A of the *T. dicoccoides* genome were found to correspond to the *PROG1* gene, whereas the chromosomes 3A, 3B, 4B, 5A and 5B have been identified on the Svevo genome, in all cases with a putative function of a zinc finger protein. Only two matches have been found by blasting the *LAZY1* gene sequence, in particular on chromosomes 6A and 6B in both genomes, being that on 6B of the durum wheat genome annotated as *LAZY1* protein. A similar result has been obtained for the *TAC1* gene, for which two transcripts have been identified in wheat on homoeologous chromosomes 5A and 5B with different annotations in the two reference genomes: a vacuolar sorting factor has been identified in the Zavitan genome whereas a NAD-dependent protein deacetylase HST1-like has been annotated in the Svevo assembly. On the contrary, different matches on all chromosomes of the A and B genomes have been obtained by using the rice *SD1* gene sequence. In all cases a putative function of 2-oxoglutarate (2OG) and Fe(II)-dependent oxygenase and gibberellin 20 oxidase 2 have been found, with some exceptions in both genomes as reported in Appendix A.

The correspondence between the physical positions of the orthologous candidates, and those of the MTAs identified in this study was then investigated. As regards the *LAZY1* gene, the transcripts TRIDC6AG060360 and TRITD6Av1G224570, respectively identified in the Zavitan and Svevo genomes, were found to be located in the same CI of the MTA D1076422 (chromosome 6A). Interestingly, a gene annotated as *LAZY1* protein was found to correspond to TRITD6Av1G224570 in the durum wheat genome. Looking at the tBLASTn results by using the *SD1* gene sequence as query, the transcript TRIDC2AG072900 from the wild accession Zavitan was found in common with those identified in the CI of the MTA S1133336 mapped on chromosome 2A, and a second region was found to correspond to this gene on 4BS where the MTA D1110414 is located, based on transcripts TRIDC4BG000680, TRIDC4BG000760 and TRIDC4BG001380. In both cases, the putative function of 2-oxoglutarate (2OG) and Fe(II)-dependent oxygenase was found. Correspondences from the durum wheat genome were instead based on TRITD2Av1G277680, TRITD7Av1G012460, TRITD7Av1G012600 and TRITD7Av1G013480, located in the same CIs of the MTAs identified on chromosome 2 and 7 of the A genome. For the B genome, a match on 4BS (TRITD4Bv1G001050), corresponding to the *SD1* gene have been retrieved in the same CIs where the MTA D1110414 for growth habit was located. Finally, regarding *TAC1* a clear correspondence was not found on both genomes, whereas the *PROG1* gene produced a match on 4B of the durum wheat genome around 650 Mb (Table 3; Appendix A). This region could correspond to the MTA identified on the same chromosome in this work, even if a very short CI has been obtained for it, as many markers resulted unmapped according the wheat consensus map available on the Triticarte website, and it was not possible to project them on the genomes. Thus, in some cases, when a match was obtained on the same chromosome, we cannot *a priori* exclude a correspondence between the genes annotated in the CIs of the MTAs identified in this work and the rice candidate genes.

## 3. Discussion

The deceleration in the relative rate of increase in yields coupled with ongoing climate change and the increase in global population represents a serious challenge for wheat breeders. It is therefore necessary to explore new gene/alleles for altering plant architecture of wheat, among other traits, to break the productivity barrier and counteract the effect of climate changes in terms of scarcity of resources. During tillering stage, prostrate growth habit with a wide tiller angle, could improve the competitive ability of the main crop against weeds and reduce the percentage of water lost by soil evaporation due to better coverage, thus improving the water use efficiency [48]. 

In the present study we reported the genetic dissection of the prostrate/erect growth habit in a panel of 184 durum wheat genotypes, including 170 winter and 14 spring types. These genetic materials were phenotyped in different environmental conditions for average temperatures, and day-length, due to latitude and sowing dates, to identify the association between some genetic markers and the expression of tiller angle. However, the expression of the trait varied by G × E although at lesser extended than G. High heritability values have been observed for this trait thus contributing to the success for the QTL detection. A normal distribution for this trait and a high broad-sense heritability (80%) was also reported in chickpea [49]. Indeed, a marker-trait association analysis has been carried out and a large number of MTAs have been identified, although few common in all environments. No common MTAs were found between the autumn sowing field trials conducted at Foggia (FF15 and FF16) and Hohenheim (HF14), probably due to the large difference in the average temperature values of the two locations and, consequently a different exposure to cold conditions that conditioned significantly the expression of prostrate growth habit [50].

As expected, the 14 genotypes identified a priori as spring types showed plant growth habits ranging from 1 (erect) to 5 (intermediate) with a mean value lower than winter durum wheat genotypes, suggesting a higher frequency of the prostrate growth habit in winter types. However, the population structure analysis divided 184 genotypes into four groups (G1, G2, G3, and G4) and spring types were clustered into group G3, showing no clear separation of winter and spring types, probably due to the low number of spring genotypes considered in the present study and/or the lack of exchange of diversity between spring and winter types in durum wheat, as previously suggested by Sieber et al. [42]. The phenotypic differences between the 4-group means were statistically significant, even excluding the spring types, suggesting that different classes for plant growth habit are not clear-cut, as there was a complete series of types from prostrate to erect depending upon temperature, length of day, and date of sowing. 

Until today few studies have been conducted on bread wheat for this character and they were also difficult to compare with the present results. For example, Li et al. [20] centered to the *Gli-A2* gliadin locus and associated to a QTL affecting prostrate growth trait. No common markers were available between the consensus wheat map and that reported by those authors, thus we cannot compare exactly the map position of the QTL. Nevertheless, the *Gli-A2* locus is known to be located on the short arm of 6A whereas we report the MTA D1076422 on the long arm of chromosome 6A (188.7 cM). Therefore, it seems located in a different region. In addition, QTLs for tiller angle have been reported, as associated to sharp eyespot resistance in wheat [51]. Out of them, one was localized on 4AL but no common markers have been found to verify the coincidence of the regions. 

More recently, Giraldo et al. [22] evaluated a collection comprising genotypes belonging to three tetraploid wheat subspecies (*durum*, *turgidum* and *dicoccum*) for different agronomic traits, including the juvenile growth habit. Three classes have been reported (prostrate, intermediate and erect) and two DArT markers, wPt-6509 and wPt-1151, located on chromosome 3A and 3B, respectively, have been found to be associated to the trait. The map position of these two markers has been compared with that reported for the MTAs identified on the same chromosomes, based on the wheat consensus map available on Triticarte website. The marker wPt-6509 was absent on the wheat consensus map, therefore we used for comparison very close markers (wPt-8203 and wPt-8876), as reported by Jing et al. [52]. These markers were located at around 260 cM on the chromosome 3A of the consensus map in a different region where the MTA D1271842 (2.7 cM) here identified, was mapped. The same was for the marker wPt-1151 located at 292.9 cM on the 3BL of the consensus map, in a different region with respect to the MTAs identified on the same chromosome. Finally, three regions (3B, 4AL and 6BS) have been reported in bread wheat by Liu et al. [21] but no marker information was available in order to confirm our results. 

The association mapping results herein obtained were compared with previous studies in which the map location of known vernalization and photoperiod genes were reported. The location of MTAs on chromosome 2B in the proximity of *Ppd-B1* gene and on chromosomes 4B, 5A and 7A that harbor vernalization genes (*Vrn-B2*, *Vrn-A1*, *Vrn-A3*, respectively), demonstrated that the sensitivity to vernalization and photoperiod response are actually not only a key components of spring/winter growth habit, but they could also play an important role in the expression of the prostrate/erect trait. However, we also found MTAs for prostrate/erect growth habit in different chromosome regions as 4A, 4B, 6A and 6B delaying sowing, under long day conditions and without satisfying the vernalization requirement of the plants. 

Interestingly, the same MTAs associated to the trait in different growing seasons and environmental growth conditions have been identified, such as the MTA D1202558 located on chromosome 2B that has been identified in FF14 in spring sowing and in FF15 experiment, in autumn sowing. A second example was represented by the QTL region identified on chromosome 5A, which explained the phenotype in different growing seasons (FF14, spring sowing, no vernalization, and HF14, autumn sowing) and in environments with very different climatic conditions (HF14, and FF14, contrastingly cold and warm, respectively), thus suggesting either divergent functions for the *VRN-A1* gene, vernalization-dependent for flowering, and vernalization-independent for tiller angle, or the presence of tightly linked different causal genes. 

The prostrate growth habit was best expressed in environments where the average temperature was lower, probably because under these conditions the vernalization requirement of genetic materials was better satisfied and the prostrate plants are less exposed to winter frosts, as shown in Hohenheim (HF14 *r* = −0.61 *p* ≤ 0.001) (Appendix A). Voss-Fels et al. (2018) demonstrated that *VRN-A1* and *VRN-H1* modulated root architecture in wheat and barley. In particular, the presence of the winter alleles consistently reduced root angle at all growth stages under greenhouse and field conditions. Therefore, photoperiod and vernalization genes could also directly contribute to the juvenile growth habit. However, based on these results, also in case of pleiotropy, they would not be the only genes involved in determining this phenotype. Similarly, a barley genome-wide association mapping study aimed to identify loci determining juvenile growth habit has resulted in significant associations with SNPs close to these determinant genes, but not the genes themselves, thus supporting tight linkage more than pleiotropy [53].

From a positional approach, candidate genes have been proposed for most of the markers included in the confidence intervals of the MTAs, many of which selected for their putative function in plant growth and development, and similar role in other species. Many zinc finger proteins, in particular C2H2- and CCCH-zinc finger proteins, have been identified, confirming the role of these family of transcription factors in determining the prostrate/erect growth habit, as reported in rice [15]. In addition, four candidate genes, such as major intrinsic protein, ankyrin repeat domain containing protein, ABC transporter, sucrose non-fermenting protein and B3 transcription factor, exhibiting a strong association with plant growth habit in our study were also validated in chickpea [49]. The gene encoding a ATP-binding cassette (ABC) transporter has been reported as involved in synchronizing plant growth with environmental and developmental changes [54]. The B3 transcription factor family is known to be involved in growth and development, in addition to flowering and vernalization responses in crop plants [55]. Indeed, we found these putative functions related to MTAs, in the same confidence intervals where also other putative candidates were positioned, such as C2H2 zinc finger proteins, and genes affecting growth regulators have been identified. Finally, the gene encoding the ankyrin repeat domain-containing protein, known to have a role in plant morphogenesis and architecture by modulating meristematic activity of shoot apical meristem [56], has been found to be associated to MTAs for the tiller angle. Genes involved in metabolism, and disease resistance proteins have been also identified in our study, confirming results from a mass spectrometry study carried out in barley in which many proteins involved in metabolism and disease/defense-related processes have been reported as influencing the juvenile growth habit [57]. 

In the search for orthologs, we also confirmed that the rice genes *PROG1*, *LAZY1*, *TAC1* and *SD1* (in turn ortholog of the barley semi-dwarf *sdw1*/*denso* gene), known to be regulators of tiller angle, are good candidates for this trait also in durum wheat. In fact, all these genes known to be regulators of the tiller angle in rice and/or in barley fall into the functional categories of the positional candidate genes identified in the present study. *PROG1* encodes a C2H2 zinc-finger protein [15], *LAZY1* plays a negative role in polar auxin transport [14,58], and the *SD1* gene with the ortholog of the *sdw1*/*denso* in barley encodes a gibberellic acid (GA)-20 oxidase enzyme [59,60], which is involved in gibberellin biosynthesis.

To the best of our knowledge, this work represents the first attempt to dissect the genetic basis of tiller angle in durum wheat by means of a genome-wide association mapping approach and could provide relevant details to select new durum wheat varieties with a plant architecture useful for improving yield under future resource-limited, agronomic and climate change scenarios. The high-quality reference sequence of the modern durum wheat cultivar Svevo [61] will also contribute to give more insight in the elucidation of the mechanisms controlling the juvenile growth habit by further studies of the candidate genes. In this new context, TILLING (Targeting Induced Local Lesions in Genomes) could represent a promising approach for gene validation studies, and for exploring the phenotypic role of the candidate genes identified in this study to identify new haplotypes controlling prostrate/erect growth habit and develop allele-specific markers for marker-assisted selection. 

## 4. Materials and Methods 

### 4.1. Plant Material

The panel of 184 durum wheat genotypes used in this study was characterized by different geographical origins, breeding history and year of release. This panel was obtained from the State Plant Breeding Institute, University of Hohenheim, Stuttgart, Germany. Among these genotypes 170 were winter durum wheat, and the remaining 14 were spring types. The collection has been previously genotyped with DArT-seq markers, being polymorphic for 30,611 markers, as described by Sieber et al. [42]. The wheat consensus map available on the Triticarte website was considered to retrieve the genetic position of these markers. 

### 4.2. Field and Greenhouse Trials and Phenotyping

The winter durum wheat panel was planted in three experimental field station: (i) at Foggia, southern Italy (41°27′44.9″ N 15°30′03.9″ E), during three consecutive growing seasons (2014–2016), different growth conditions (field trial and greenhouse) and two sowing periods (autumn and spring). These experimental conditions were designed as FF14 (field trial with spring sowing on 3 April 2014 in small plots not replicated), FF15 (field trial carried out during 2014–2015 with autumn sowing on 15 December 2014 in plots with three replications), FG15 (Greenhouse experiment with winter sowing on 7 March 2015 in small plots with two replications) and FF16 (field trial carried out during 2015–2016 with autumn sowing on 7 December 2015 in plots with three replications); (ii) at Probstdorf, eastern Austria (48°10′13.4″ N 16°36′57.0″ E) during 2013–2014 growing season (PF13) in a field trial with autumn sowing and three replications; (iii) at Hohenheim, southern Germany (48°42′42.2″ N 9°12′42.5″ E), during the growing season 2013–2014 (HF14) in field trial with no replications. The three experimental sites are characterized by contrasting climatic conditions since Foggia is characterized by a typical Mediterranean climate with mild winters and dry summers whereas Probstdorf and Hohenheim in Austria and Germany are characterized by a continental climate with cold winters (average daily temperatures around 0 °C), and mild summers with the maximum temperatures around 22/24 °C in July and August. Late sowing carried out at Foggia during spring seasons (FF14 and FG15) were designed to evaluate the expression of juvenile growth habit trait excluding any natural vernalization. 

The juvenile growth habit trait was estimated by measuring the tiller angle between the last developed tillers and the ground level with a protractor at the maximum tillering stage (GS25 to GS29 according to Zadoks et al. [62]) by following UPOV [63] guidelines: 1, erect; 3, semi-erect; 5, intermediate; 7, semi-prostrate; 9, prostrate. Figure 5 showed the extreme phenotypes (erect and prostrate, respectively) of two genotypes as an example. In order to avoid confusion with seasonal growth habit (winter, facultative and spring type), in this paper growth habit was referred exclusively to the aptitude of the leaves and tillers to form an angle of different amplitude respect to an imaginary middle axis at tillering stage. Frost tolerance data, visually scored on a scale from 1 (no damage) to 9 (no plant survived) from the field trial conducted at Hohenheim, were used to investigate the relationships between with juvenile growth habit loci and the *Vrn* locus on chromosome 5A. 

### 4.3. Statistical Analyses

Descriptive statistics were computed. The probability density function (PDF) of the habit by environment (i.e., location × year), mean of genotypes across sites and genotype within sites were built. Phenotypic data were analyzed using MIXED Model Equation (MME, SAS/STAT 9.2, SAS Institute Inc., Cary, NC, USA) with Residual Maximum Likelihood (REML) estimation method and growing degree days (Tb = 0 °C) as a covariate. No data transformation was performed since MME can handle non-normal data and correct for heteroscedasticity [64]. Best Linear Unbiased Estimation (BLUE) and Best Linear Unbiased Predictions (BLUPs) were computed to take into account the genotype by environment interactions (GxE), according to the recommendation by Piepho et al. [65]. Broad sense heritability (h^2^) was computed by solving both the MME and the General Linear Model (GLM) according to [66]. Pearson’s correlations for evaluated traits among different environments were statistically analyzed at the 0.05 probability level.

### 4.4. Population Structure and GWA Analyses

The population structure was investigated using the model-based clustering method as implemented in Structure 2.3.4 [67] and the Discriminant Analysis of Principal Components (DAPC) analysis implemented in the Adegenet package for the R software (3.5.3, Version: Great Truth) [68]. We used an admixture model within the first method using the options ‘correlated allele frequencies among populations’ and ‘infer the degree of admixture by the data’. For each K (number of hypothetical populations), 20 runs (burn-in length, 100,000; iterations, 100,000) were carried out, and the most likely number of K was determined using the method from Evanno et al. [69] as implemented in the online program STRUCTURE Harvester [70]. Both single-*locus* and multi-*locus* GWA analyses were performed on the 184 durum wheat genotypes using 30,475 DArT-seq markers. The prostrate/erect growth habit at the tillering stage was analysed for each single environment, as also the LSmeans obtained from across all environments. Specifically, the general linear model (GLM) and the mixed linear model (MLM) were used as implemented in GAPIT (http://zzlab.net/GAPIT; [71,72,73]). Then, the modified multi-*locus* mixed model called Fixed and random model Circulating Probability Unification (FarmCPU) (http://zzlab.net/FarmCPU; [74]) was also used for association analysis. The multi-*locus* methods are often used when a complex trait is under study to disentangle the role of many *loci* showing significant effects in the expression of a phenotype [75,76]. A multi-*locus* approach aims at enhancing the false-discovery rate and the QTL detection power by incorporating one or several markers as cofactors in a stepwise MLM, thus removing the confounding effect between testing markers and kinship [74]. Results from the different methods were compared using Quantile-Quantile (Q-Q) plots and those from FarmCPU were chosen for further discussion. The outputs obtained from GWAS by single-*locus* MLM were additionally presented relatively to Hohenheim, where in addition to plant growth habit, frost tolerance data were also available. Standard Bonferroni correction (*p* ≤ 0.05) were chosen to evaluate the threshold for significant associations (*p* = 1.64E-06) and relevant MTAs were identified accordingly. Figures for GWAS results were drawn using the “A Memory-efficient, Visualization-enhanced, and Parallel-accelerated Tool for Genome-Wide Association Study” (MVP) package (https://github.com/XiaoleiLiuBio/rMVP). 

### 4.5. Identification of Candidate Genes 

The sequence of the significant markers identified by association mapping, as well as markers included in their CI, calculated according to the LD decay estimated by Sieber et al. [42] on the same winter durum wheat population and using the same SNP markers, were used to find positional candidate genes. The left and the right markers of the interval, together with the tagging markers when possible were based on sequence availability, and some internal markers were projected to the genome assemblies of the *T. dicoccoides* accession Zavitan, and of the durum wheat cultivar Svevo by a BLASTn search against their gene sets (threshold E-10) [61,77], in order to identify, by imposing functional hypotheses, candidate genes for prostrate/erect growth habit trait. All genes comprised in the LD decay intervals were retrieved with their functional annotations in the corresponding wild emmer and durum wheat genome intervals for further discussion.

### 4.6. Search for Orthologous of the Rice Genes PROG1, TAC1, LAZY1, SD1 in Wheat

The sequence of the *PROG1* gene isolated in rice was retrieved by Tan et al. [15] whereas *TAC1* (LOC_Os09g35980) and *LAZY1* (LOC_Os11g29840) sequences were searched on the rice genome database available at http://rice.plantbiology.msu.edu/ website. A barley GA 20-oxidase gene (Hv20ox2) had been proposed as a candidate for *sdw1*/*denso*, likely ortholog to rice *sd1*/*Os20ox2* gene [23,59,60]. For this reason, also *SD1* (LOC_Os01g0883800) sequence was searched on the rice genome database. The rice protein sequences were used for a tBLASTn search against the genomes of the wild emmer wheat accession Zavitan and of the durum cultivar Svevo (threshold E-10). The physical map position of the best sequence hits was then compared to that of the MTAs identified in their LD decay intervals, to investigate the correspondence.

## Figures and Tables

**Figure 1 ijms-21-00394-f001:**
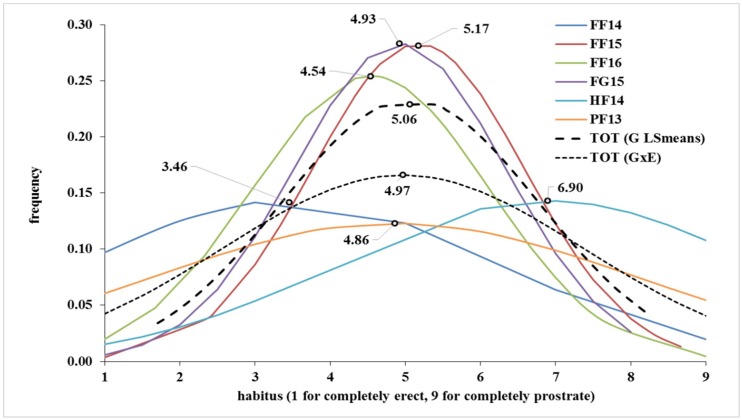
Probability density function (PDF) of the growth habit in the environments under study. TOT for total distribution by means of genotypic means across sites (G LSmeans, i.e., least squared means of a given genotype across environments) and genotypic values in all environments (G × E) mean value of each genotype in each environment). Open points indicate the mean of each distribution.

**Figure 2 ijms-21-00394-f002:**
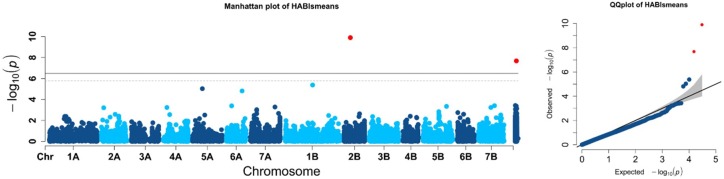
Manhattan plot showing the chromosome location of significant marker-trait associations for prostrate growth habit inputted as the LSmeans across the six environments. Significant MTAs are highlighted in red (*p* ≤ 0.01).

**Figure 3 ijms-21-00394-f003:**
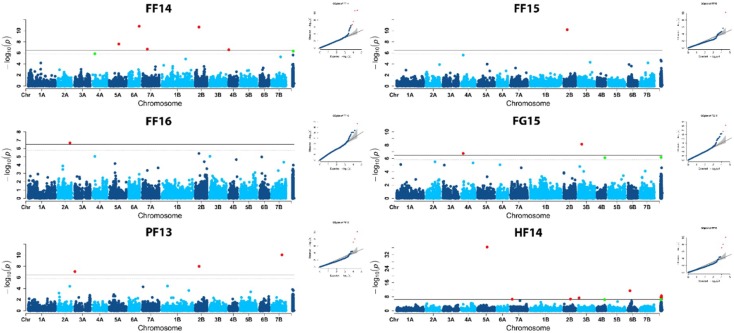
Manhattan plots showing the chromosome location of significant marker-trait associations for prostrate growth habit from FarmCPU analysis. Significant MTAs are highlighted in red (*p* ≤ 0.01) and green (*p* ≤ 0.05).

**Figure 4 ijms-21-00394-f004:**
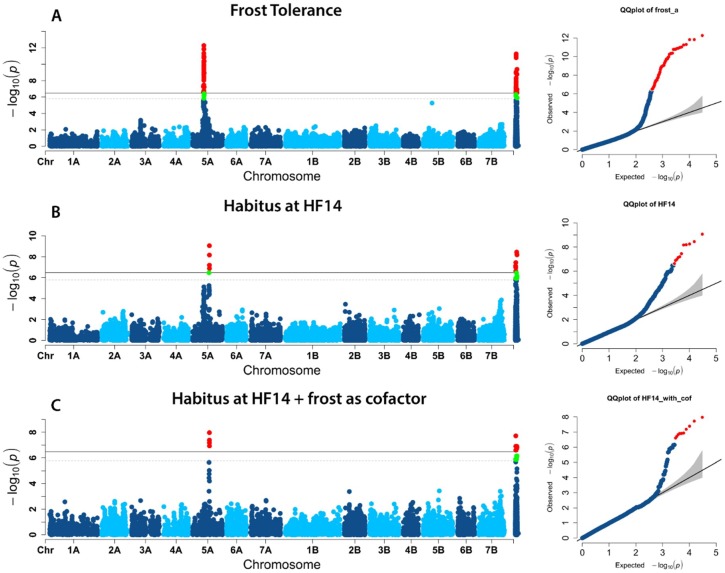
Manhattan plots showing the MLM analysis (using population structure and kinship as covariates) on prostrate growth habit (**A**) and frost tolerance (**B**). MLM analysis was also performed on growth habit adding the frost tolerance as a cofactor (**C**). Significant MTAs are highlighted in red (*p* ≤ 0.01) and green (*p* ≤ 0.05).

**Figure 5 ijms-21-00394-f005:**
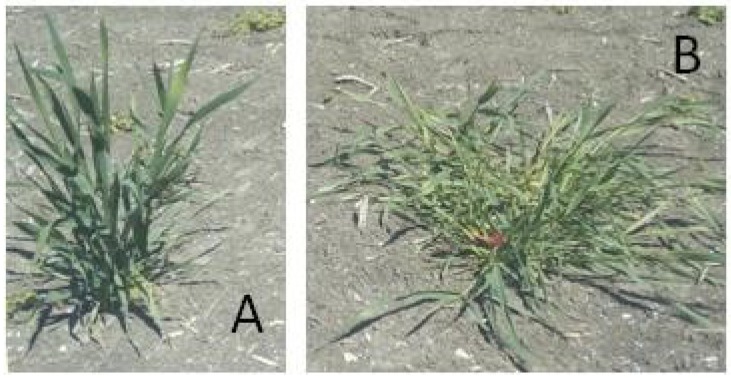
Examples of two genotypes of the collection used in this study, characterized by a complete erect (**A**) and prostrate (**B**) habit.

**Table 1 ijms-21-00394-t001:** Descriptive statistics of the prostrate/erect growth habit (1 for completely erect, 9 for completely prostrate) in the environments under study.

	FF14	FF15	FF16	FG15	HF14	PF13	TOT (G LSmeans)	TOT (G × E)
Mean	3.46	5.17	4.54	4.93	6.90	4.86	5.06	4.97
Standard Error	0.25	0.10	0.12	0.11	0.25	0.27	0.13	0.08
Coefficient of Variation	80.4	27.3	34.5	28.6	40.4	66.8	33.9	48.4
σ^2^_G_	n.a.	5.05	7.35	3.65	15.55	14.02	n.a.	30.29
F	n.a.	4.05	16.26	18.46	45.38	9.94	n.a.	10.69
P	n.a.	<0001	<0001	<0001	<0001	<0001	n.a.	<0001
Kurtosis	−1.01	−0.37	−0.43	−0.08	−0.12	−1.78	−1.40	−0.93
Skewness	0.61	0.06	−0.46	−0.21	−1.28	0.08	−0.28	−0.06
Min	1.0	1.0	1.0	1.0	1.0	1.0	1.7	1.0
Max	9.0	8.7	9.0	8.0	9.0	9.0	8.2	9.0
Range	8.0	7.7	8.0	7.0	8.0	8.0	6.5	8.0
h^2^_B_ with covariance structure (Mixed Model)	n.a.	0.997	0.998	0.999	0.999	0.991	n.a.	0.910
h^2^_B_ with fixed effects (GLM)	n.a.	0.941	0.903	0.999	0.978	0.793	n.a.	0.652

n.a, not applicable; FF14, Foggia 2013–2014; FF15, Foggia 2014–2015; FF16, Foggia 2015–2016; FG15, Foggia greenhouse experiment 2015; HF14, Hohenheim 2014–2015; PF13, Probstdorf 2013–2014; TOT for total distribution by means of genotypic means across sites (G LSmeans, i.e., least squared means of the genotype) and genotypic values in all environments (G × E).

**Table 2 ijms-21-00394-t002:** Results of analysis of variance with type 3 error [degrees of freedom (df), Habit mean squares (MS) and *p*] and Covariance parameter estimate (Cov) for the computation of the h^2^ across environments from the mixed model when including all treatments in the covariance structure for growth habit (1 for completely erect, 9 for completely prostrate) across environments.

Source of Variation	df	Habit MS	*p*	Cov
Environment (E)	5	92.005	<0001	0.315
Blocks within Environment	7	2.080	0.0048	0.010
Genotype (G)	182	30.287	<0001	3.124
G × E	899	3.082	<0001	1.191
Error	1229	0.708	-	0.708

**Table 3 ijms-21-00394-t003:** Summary of the MTAs identified with FarnCPU.

Marker	Environment	Chr	cM	*p* Value	PEV *	Literature
S1133336	FF16	2A	217.7	2.22 × 10^−07^	0.15	
**D1202558**	**FF14**	**2B**	**62.3**	**2.15 × 10^−11^**	**0.27**	***Ppd-B1***
	**FF15**	**2B**	**62.3**	**6.30 × 10^−11^**	**0.10**	***Ppd-B1***
	**LSmeans**	**2B**	**62.3**	**1.26 × 10^−10^**	**0.16**	***Ppd-B1***
D2294169	PF13	2B	65.1	9.91 × 10^−09^	0.12	*Ppd-B1*
D1137224	HF14	2B	120.3	2.07 × 10^−07^	0.05	
D1271842	PF13	3A	2.7	8.54 × 10^−08^	0.11	
D1266232	HF14	3B	23.9	4.73 × 10^−08^	0.04	
S1049173	FG15	3B	71.8	7.11 × 10^−09^	0.07	
D1665929	FF14	4A	37.2	1.39 × 10^−06^	0.24	
	FG15	4A	37.2	1.80 × 10^−07^	0.06	
D1110414	FF14	4B	0	2.77 × 10^−07^	0.25	
D1395268	HF14	4B	133.6	4.83 × 10^−07^	0.01	*Vrn-B2*
D1720107	FG15	4B	138.4	7.89 × 10^−07^	0.07	*Vrn-B2*
D2276320	HF14	5A	167.7	4.51 × 10^−07^	0.51	*Vrn-A1*
D1721703	FF14	5A	168.6	2.48 × 10^−08^	0.03	*Vrn-A1*
D1076422	FF14	6A	188.7	1.57 × 10^−11^	0.20	
D2289020	HF14	6B	36.8	3.92 × 10^−12^	0.22	
D2295851	HF14	7A	32.3	2.74 × 10^−07^	0.08	
D1031337	FF14	7A	91.8	2.05 × 10^−07^	0.22	*Vrn-A3*
D1112046	PF13	7B	184.4	8.66 × 10^−11^	0.08	
D4004513	HF14	unmapped		5.56 × 10^−07^	0.18	
D1744736	HF14	unmapped		2.28 × 10^−08^	0.19	
D3944539	FG15	unmapped		6.76 × 10^−07^	0.04	
D3946194	HF14	unmapped		3.54 × 10^−07^	0.18	
**D2277949**	**LSmeans**	**unmapped**		**2.07 × 10^−08^**	**0.21**	
D3935715	HF14	unmapped		2.60 × 10^−09^	0.12	
D3533805	HF14	unmapped		2.74 × 10^−07^	0.08	
S984195	FF14	unmapped		4.83 × 10^−07^	0.03	
S1218298	HF14	unmapped		2.50 × 10^−08^	0.10	

* PEV = Proportion of Explained Variance. Considering that Circulating Probability Unification (FarmCPU) does not provide the proportion of explained phenotypic variance for each MTA, we here report the adjusted R^2^ values for each significant SNP that were calculated using the lm() function in R. In bold, the SNP markers associated with the prostrate/erect growth habit in all environments through the calculation of the LSmeans.

**Table 4 ijms-21-00394-t004:** Size and genes content of the physical regions retrieved from the Zavitan genome corresponding to the CIs of the MTAs identified.

MTA	chr	CI (cM)	CI Start (bp)	CI End (bp)	CI (Mbp)	Number of Annotated Genes	Number of Related-Growth Habit Genes
S1133336	2A	217.5–219.7	734724817	741838496	7.1	97	7
D1202558	2B	60.3–64.7	45229865	51452747	6.2	38	3
D1137224	2B	117.7–124.3	117320738	149930630	32.6	73	5
D1271842	3A	0.6–6.5	2640850	15387176	12.7	58	1
D1266232	3B	19.7–29.4	4996463	17173269	12.2	58	4
S1049173	3B	68.2–75.3	45389928	63515801	18.1	49	3
D1665929	4A	37.1–39.8	47676808	77735733	30	107	7
D1110414	4B	0–3	2180245	13531313	11.3	113	9
D1395268	4B	132.4–138	657599661	661590000	5.6	44	2
D2276320	5A	164.3–168.9	575035656	590239189	15.2	140	14
D1076422	6A	185.2–191.1	607923652	618547327	10.6	129	12
D2289020	6B	35.5–36.8	32844660	47681687	14.8	104	5
D2295851	7A	31.4–38	15519984	23593672	8.1	35	2
D1031337	7A	91.2–92.4	50882080	56636443	5.7	57	6
D1112046	7B	181.9–188.8	680402861	698138699	17.7	35	3

**Table 5 ijms-21-00394-t005:** Size and genes content of the physical regions retrieved from the durum wheat Svevo genome corresponding to the CIs of the MTAs identified.

MTA	chr	CI (cM)	CI Start (bp)	CI End (bp)	CI (Mbp)	Number of Annotated Genes	Number of Related-Growth Habit Genes
S1133336	2A	217.5–219.7	739004624	746220682	7.2	202	10
D1202558	2B	60.3–64.7	44096854	49732398	5.6	148	7
D1137224	2B	117.7–124.3	109991884	143184509	33.2	358	20
D1271842	3A	0.6–6.5	963147	7004107	6	196	5
D1266232	3B	19.7–29.4	4675018	15213712	10.5	235	13
S1049173	3B	68.2–75.3	49798003	61284099	11.5	178	9
D1665929	4A	37.1–39.8	44642299	76324355	31.7	461	10
D1110414	4B	0–3	1729305	13558565	11.8	281	11
D1395268	4B	132.4–138	658918900	659539176	0.6	33	1
D2276320	5A	164.3–168.9	532806577	556694196	23.9	419	14
D1076422	6A	185.2–191.1	602368652	614383914	12	325	17
D2289020	6B	35.5–36.8	30090166	54319106	24.2	319	13
D2295851	7A	31.4–38	18289584	27148666	8.8	240	9
D1031337	7A	91.2–92.4	55718952	60630163	4.9	127	5
D1112046	7B	181.9–188.8	657142046	679568205	22.4	326	10

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
