# Peer review of "Genome-Wide Association Mapping of Prostrate/Erect Growth Habit in Winter Durum Wheat"

_ijms, 2020, doi:10.3390/ijms21020394_

Round 1

Reviewer 1 Report

The study used a panel of 184 durum wheat cultivars to map QTLs associated with growth habit. A number of marker-trait associations were identified, however only 2 potential QTLs were identified across all environments. Based on the GWAS results, candidate genes were assessed, based on previous studies and genomic location.

The potential candidate genes were discussed in the context of other studies of these genes.

Overall, the approaches used and results obtained in the study are clearly presented. The discussion is concentrated on the candidate genes and comparisons to other studies.

The growth habit trait was shown to have high heritability in all environments. However, very few MTAs were shared across environments, and those that were found in two environments were with differing conditions – e.g. spring vs autumn sowing, field vs greenhouse. Some discussion about this could be included in the manuscript, e.g. why no common MTAs were found between the two autumn sowings (apart from the MTAs common to all environments).

Author Response

Below is a detailed response to the referee#1 and an electronic copy of our revised manuscript is attached. We have incorporated all suggested changes in RED.

Response to Reviewer 1 Comments

Point 1: The growth habit trait was shown to have high heritability in all environments. However, very few MTAs were shared across environments, and those that were found in two environments were with differing conditions – e.g. spring vs autumn sowing, field vs greenhouse. Some discussion about this could be included in the manuscript, e.g. why no common MTAs were found between the two autumn sowings (apart from the MTAs common to all environments).

Response 1: Thanks for the suggestion, we have inserted a comment in the text (L331-L336)

Reviewer 2 Report

The authors performed GWAS of prostrate/erect habit using durum wheat cultivars. The GWAS identified some MTAs and give the understandings of prostrate/erect growth habit in durum wheat. It is worth publishing. Before publication, I would like to confirm some points and recommend revisions as listed below.

Major comments

If the authors suggested that vrn-1 could contribute to the juvenile growth habit, why did the authors not discuss the difference of prostrate/erect growth habit between spring and winter wheat?

And I am interested in whether there is difference of prostrate/erect growth habit among the four groups identified the STRUCTURE analysis. I think this can give better understandings of GWAS for readers since population stratification can produce false positive associations.

The authors investigated the location of MTAs based on the LD. Because the DArT markers have tag sequences, we can know the genomic positions of the DArT markers. Why did you choose the LD instead of genome sequences?

Minor comments

L72-L73. The name of genes should be italic (e.g. LA1)

L130 “Genotypic”? Do you mean “Phenotypic”?

L134 “P≤0.001” should be changed to “P ≤ 0.001”.

L162 “χ2=386.001, P<0.0001” should be changed to “χ2 = 386.001, P < 0.0001”.

L203 How did you calculate the LD decay of 5cM? From the other data or your data?

L282-L328 This part should be moved to the “Results” section. And then “Discussion” should be reconsidered.

L319 Do you mean “Results from GWAS using MLM”?

L319 Is the GWAS analysis of frost resistance and tiller angle form this study? There is no description about this in M&M.

L320 “p=5.34E-13” should be changed to “p = 5.34E-13”.

L322 and L323 “p=8.68E-10” and “p=7.54E-06” should be changed to “p = 8.68E-10” and “p = 7.54E-06”, respectively.

L331 and 334 “Gli-A2” should be italic and “locus” should not be italic.

L368 “VRN-A1” should be italic.

L374 “P≤0.001” should be changed to “P ≤ 0.001”.

L380 “loci” should not be italic.

L465 “loci” and “locus” should not be italic.

L504 “p≤0.05” should be changed to “p ≤ 0.05”.

L505 “p=1.64E-06” should be changed to “p = 1.64E-06”.

Figure 5 I did not found A, B and C. The authors should change (a), (b) and (c) to (A), (B) and (C), respectively. And “P ≤0. 01” should be replaced with “P ≤ 0. 01”.

Author Response

Below is a detailed response to the referee#2 and an electronic copy of our revised manuscript is attached. We have incorporated all suggested changes in RED.

Response to Reviewer 2 Comments

Major comments

Point 1: If the authors suggested that vrn-1 could contribute to the juvenile growth habit, why did the authors not discuss the difference of prostrate/erect growth habit between spring and winter wheat?

Response 1: Thank you for your note, we have inserted a comment in the text (L337-L339)

Point 2: And I am interested in whether there is difference of prostrate/erect growth habit among the four groups identified the STRUCTURE analysis. I think this can give better understandings of GWAS for readers since population stratification can produce false positive associations.

Response 2: Following the previous point we added a comment in the text (L160-L162 and L339-L347).

Point 3: The authors investigated the location of MTAs based on the LD. Because the DArT markers have tag sequences, we can know the genomic positions of the DArT markers. Why did you choose the LD instead of genome sequences?

Response 3: It is true, the DArT markers have tag sequences, but they are not all available on the Triticarte website. For 7 of the 9 unmapped SNPs the sequence was not available while for the remaining two SNPs (D3935715 and D3533805) the sequences were too short to produce a reliable result. For this reason, we have used LD.

Minor comments

Point 4: L72-L73. The name of genes should be italic (e.g. LA1)

Response 4: Done L72

Point 5: L130 “Genotypic”? Do you mean “Phenotypic”?

Response 5: Thank you for your note. We replaced the term. L130

Point 6: L134 “P≤0.001” should be changed to “P ≤ 0.001”.

Response 6: Done L134

Point 7: L162 “χ2=386.001, P<0.0001” should be changed to “χ2 = 386.001, P < 0.0001”.

Response 7: Done L164

Point 8: L203 How did you calculate the LD decay of 5cM? From the other data or your data?

Response 8: The LD was calculated by Siebier et al., 2015 on the same winter durum wheat population and using the same SNP markers, as specified in the paragraph MM (L527-L528)

Point 9: L282-L328 This part should be moved to the “Results” section. And then “Discussion” should be reconsidered.

Response 9: Moved, now L190-L238

Point 10: L319 Do you mean “Results from GWAS using MLM”?

Response 10: Yes, we corrected as suggested L228

Point 11: L319 Is the GWAS analysis of frost resistance and tiller angle form this study? There is no description about this in M&M.

Response 11: Yes, the GWA analysis for frost tolerance was conducted in the present study, as indicated in MM L519-L521.

Point 12: L320 “p=5.34E-13” should be changed to “p = 5.34E-13”.

Response 12: Done L229

Point 13: L322 and L323 “p=8.68E-10” and “p=7.54E-06” should be changed to “p = 8.68E-10” and “p = 7.54E-06”, respectively.

Response 13: Done L231 and 232

Point 14: L331 and 334 “Gli-A2” should be italic and “locus” should not be italic.

Response 14: Done L349, L350 and L352

Point 15: L368 “VRN-A1” should be italic.

Response 15: Done L386

Point 16: L374 “P≤0.001” should be changed to “P ≤ 0.001”.

Response 16: Done L 392

Point 17: L380 “loci” should not be italic.

Response 17: Done L 398

Point 18: L465 “loci” and “locus” should not be italic.

Response 18: Done L483

Point 19: L504 “p≤0.05” should be changed to “p ≤ 0.05”.

Response 19: Done L521

Point 20: L505 “p=1.64E-06” should be changed to “p = 1.64E-06”.

Response 20: Done L522

Point 21: Figure 5 I did not find A, B and C. The authors should change (a), (b) and (c) to (A), (B) and (C), respectively. And “P ≤0. 01” should be replaced with “P ≤ 0. 01”.

Response 21: Done. Please see the new Figure 4(*).

(*) In the previous version there was an error in attributing the numbers of the figures, in this new version of the manuscript we have corrected the numbering of the figures.
